# Glutamic Acid at Position 343 in PB2 Contributes to the Virulence of H1N1 Swine Influenza Virus in Mice

**DOI:** 10.3390/v17071018

**Published:** 2025-07-20

**Authors:** Yanwen Wang, Qiu Zhong, Fei Meng, Zhang Cheng, Yijie Zhang, Zuchen Song, Yali Zhang, Zijian Feng, Yujia Zhai, Yan Chen, Chuanling Qiao, Huanliang Yang

**Affiliations:** State Key Laboratory for Animal Disease Control and Prevention, Harbin Veterinary Research Institute, Chinese Academy of Agricultural Sciences, Harbin 150069, China; 15290068938@163.com (Y.W.);

**Keywords:** influenza virus, H1N1 subtype, PB2, molecular basis, pathogenicity

## Abstract

The H1N1 swine influenza viruses CQ91 and CQ445, isolated from pigs in China, exhibited distinct virulence in mice despite sharing similar genomic constellations. CQ91 demonstrated higher pathogenicity (MLD_50_: 5.4 log_10_ EID_50_) and replication efficiency in mice compared to CQ445 (MLD_50_: 6.6 log_10_ EID_50_). Through reverse genetics, we found that the attenuation of CQ445 was due to a single substitution of glutamic acid (E) with lysine (K) at position 343 in the PB2 protein. Introducing the CQ445-PB2 (343K) into CQ91 significantly reduced viral replication and pathogenicity in mice, while replacing CQ445-PB2 with CQ91-PB2 (343E) restored virulence. In vitro studies showed that the K343E mutation impaired viral replication in MDCK and A549 cells and reduced polymerase activity in minigenome assays. Mechanistically, the amino acid at position 343 in the PB2 affects the transcription stage of the viral replication process. Structural modeling indicated that the charge reversal caused by E343K altered local electrostatic interactions without major conformational changes. Phylogenetic analysis revealed that PB2-343E is highly conserved (>99.9%) in human and swine H1/H3 influenza viruses, suggesting that PB2-343E confers an adaptive advantage. This study identifies PB2-343E as a critical determinant of influenza virus pathogenicity in mammals, highlighting its role in host adaptation.

## 1. Introduction

Influenza A virus (IAV) is a single-stranded RNA virus characterized by its rapid evolutionary potential and exceptional capacity for cross-species transmission. Beyond its intrinsic genetic plasticity, IAV’s interspecies spread is facilitated by ecological niches that promote viral gene reassortment. Pigs serve as critical intermediary hosts due to their respiratory epithelial cells expressing both α-2,6-sialic acid (human-type) and α-2,3-sialic acid (avian-type) receptors, enabling coinfection by avian, swine, and human IAV strains and creating opportunities for genetic recombination. The 2009 pandemic H1N1 (2009/H1N1) strain demonstrates a high propensity for genetic reassortment with other IAVs. Subsequent to the introduction of the 2009/H1N1 virus into swine populations, worldwide monitoring has recorded numerous reassortment occurrences between this virus and endemic swine influenza strains. In continents including the Americas, Europe, and Asia, researchers have identified reassortant influenza viruses carrying gene segments from both the 2009 pandemic H1N1 virus and local swine influenza lineages [1,2,3]. This genetic reassortment not only enriches viral diversity but also enhances adaptability to different host species.

Eurasian avian-like H1N1 (EA H1N1) reassortant viruses exemplify the complexity of IAV genetic reassortment, having evolved into distinct genotypes through repeated reassortment with human and swine influenza virus lineages. Notably, genotypes G4 and G5 became widely distributed in Chinese pig populations during 2013–2019 [4]. Viruses frequently undergo critical genomic modifications that alter their interhost transmission efficiency and pathogenic potential. The identification of these genetic alterations will enhance our understanding of viral virulence determinants and facilitate the development of countermeasures. EA-reassortant viruses have acquired polymerase genes from the 2009 pandemic H1N1 virus, a genetic exchange that may give rise to novel reassortants with enhanced virulence compared to their parental strains. The influenza viral polymerase complex, composed of polymerase acidic (PA), polymerase basic 1 (PB1), and PB2 subunits, is essential in viral genome replication and transcription within the cell nucleus. Viral polymerase adaptation is critical in efficient viral replication in a new host following cross-species transmission. Notably, as of 2022, a total of 23 human infections with EA H1N1 viruses had been documented in China, including two fatal cases, indicating that these viruses have acquired the capacity to cause severe disease in humans [5,6,7,8]. The pathogenicity of influenza viruses in mammalian hosts is governed by amino acid substitutions. While numerous studies have explored the molecular basis of avian influenza virus adaptation to mammals and identified virulence-related amino acids in various viral proteins, our understanding of the pathogenic and transmissible mechanisms of EA-reassortant viruses remains limited. Our study demonstrates that EA-reassortant viruses exhibit enhanced pathogenicity and transmissibility in ferrets following the acquisition of key mutations in the PA protein [9]. It is not unexpected that multiple alterations in the polymerase genes of novel virus strains could lead to their adaptation to humans and enhanced pathogenicity. During routine surveillance, we isolated two EA H1N1 viruses, A/swine/Chongqing/91/2018 (CQ91) and A/swine/Chongqing/445/2018 (CQ445), from a slaughterhouse. Genetic analysis showed high similarity between the viruses, though they differed in replication efficiency and virulence in mice. Notably, all known influenza A virulence markers were identical in both strains, suggesting that uncharacterized amino acid residues or motifs underlie their divergent pathogenicity in the mouse model.

In this study, we employed reverse genetics to generate a series of reassortants and mutants with a CQ91 genetic background. Following virulence assessment in mice, we discovered a previously undescribed virulence-linked determinant within the PB2 protein and delved into the mechanistic basis underlying pathogenicity variations between the two viral strains in living mice.

## 2. Materials and Methods

### 2.1. Cells and Viruses

Madin–Darby canine kidney (MDCK) (obtained from ATCC) and human embryonic kidney (HEK 293T) (obtained from ATCC) cell lines were cultivated in Dulbecco’s modified Eagle’s medium (DMEM) supplemented with 10% fetal bovine serum (FBS). Human lung adenocarcinoma epithelial cell lines (A549) (obtained from ATCC) were cultivated in Ham’s F12K (Kaighn’s) medium supplemented with 10% fetal bovine serum (FBS), and all cells were maintained at 37 °C in a 5% CO_2_ incubator. The H1N1 viruses, CQ91 and CQ445, were isolated from a slaughterhouse during routine surveillance. Virus stocks were generated by propagating them in 10-day-old specific-pathogen-free (SPF) embryonated chicken eggs, then stored at −70 °C for subsequent RNA extraction and animal experiments.

### 2.2. Sequence Analysis and Construction of Plasmids for Virus Rescue

Viral RNA (vRNA) was isolated from the allantoic fluid through a standardized extraction protocol, after which it underwent reverse transcription (RT) using a commercially available kit to generate complementary DNA (cDNA). A set of fragment-specific primers was used for the PCR amplification [10]. After the complete sequencing of the two viral genomes, their amino acid sequences were compared to identify differences. The eight gene segments of CQ91 and CQ445 were inserted into the vRNA-mRNA bidirectional transcription vector pBD to rescue rCQ91, rCQ445, and all reassortant viruses as described previously [11]. To confirm the lack of unintended mutations, full-genome sequencing was performed on all rescued viruses.

### 2.3. Viral Replication in MDCK and A549 Cells

To assess virus growth kinetics, MDCK and A549 cell monolayers were infected with parental and recombinant viruses at a multiplicity of infection (MOI) of 0.01. Supernatants were harvested at 12, 24, 36, and 48 h post-infection and then preserved at –70 °C for long-term storage. Viral titers were determined by performing endpoint titration assays in MDCK cells, with results reported as the mean log_10_ TCID_50_ per milliliter (mean ± standard deviation) using the method described in reference [12]. The growth data represent the averaged results of three independent experiments.

### 2.4. Mouse Experiments

BALB/c mice (Vital River Laboratories, Beijing, China) were randomly allocated into groups of three for the initial experiment. Each mouse was anesthetized with avertin at a dose of 2.5% *w*/*v*) administered intraperitoneally to ensure deep sedation without compromising physiological functions. Following anesthesia, a 50 μL aliquot containing a 10^6.0^ 50% egg infectious dose (EID_50_) of the target virus was carefully inoculated intranasally using a micropipette, ensuring that the virus entered the respiratory tract directly. Two groups were infected with the target virus. Three days post-inoculation, mice were humanely euthanized via cervical dislocation in accordance with institutional animal care guidelines. Subsequently, for one group, tissues including the nasal turbinate, lungs, spleen, kidneys, and brain were collected from each animal under aseptic conditions. These samples were immediately stored in sterile vials containing viral transport medium and kept at −80 °C until further processing for virus titration in embryonated chicken eggs. For the other group, lungs were fixed by submersion in 10% neutral buffered formalin for hematoxylin–eosin (H&E) staining or immunohistochemical (IHC) assays with a mouse anti-NP monoclonal antibody. For the determination of the 50% mouse lethal dose (MLD_50_), separate groups of six mice were used. Serial 10-fold dilutions of the virus stock, ranging from 10^3^ to 10^7^ EID_50_ per 50 μL, were prepared using sterile phosphate-buffered saline. Each dilution was then administered intranasally to a designated group, with strict precautions taken to prevent cross-contamination between samples. Over the subsequent 14-day observation period, individual body weights and survival status were meticulously recorded daily. Infected mice exhibiting weight loss exceeding 25% of their baseline body weight were promptly euthanized following approved humane endpoints as described previously [13]. Mortality data were analyzed using the Reed–Muench method to calculate the MLD_50_ value [14], providing critical insights into viral pathogenicity.

### 2.5. Viral Minigenome Luciferase Assay

A viral minigenome luciferase assay was conducted as previously reported [9] to assess the relative polymerase activities of the H1N1 viruses. Subsequently, the PB2, PB1, PA, and NP genes were cloned into the pCAGGS protein expression vector. Mutations were introduced through a specific primer design following the manufacturer’s instructions (Vazyme, Nanjing, China). To confirm the integrity of the constructs, all plasmids underwent sequencing to exclude any unintended mutations. Using Lipofectamine LTX and Plus reagents, HEK293T cells were co-transfected with plasmids expressing PB2, PB1, PA, and NP proteins, a pPolII-Luc plasmid that drives firefly luciferase gene expression from a virus-like RNA under the control of the human RNA polymerase I promoter, and a control plasmid pRL-TK encoding Renilla luciferase. Transfected cells were incubated at 37 °C for 24 h. Following incubation, cells were lysed, and the relative luciferase activity was measured using a dual-luciferase reporter assay kit according to the Promega protocol. Luciferase activities were quantified with a GloMax 96 microplate luminometer (Promega, Madison, WI, USA). The presented data represent the mean values ± standard deviations from three independent experimental replicates.

### 2.6. Quantitative Real-Time PCR (qRT-PCR) of vRNA, cRNA, and mRNA

The absolute quantification of vRNA, cRNA, and mRNA was performed for the four viruses. When HEK293T cells reached approximately 70% confluence, plasmid transfection was initiated. Specifically, cells were double-washed with Opti-MEM serum-free medium (Invitrogen, Carlsbad, CA, USA), followed by the transfection of pCAGGS-PB1, pCAGGS-PB2 (wild-type or mutant), pCAGGS-PA, pCAGGS-NP, and pPolI-vRNA plasmids using Lipofectamine LTX Reagent with PLUS Reagent (Thermo Fisher Scientific, Waltham, MA, USA). At 24 h post-transfection (hpt), total RNA was extracted from the cells using the RNA simple Total RNA Extraction Kit (Tiangen, Beijing, China). After RNA isolation, the reverse transcription of the vRNA, cRNA, and mRNA of each virus was performed using Tag-primers (Appendix A) with the First Strand cDNA Synthesis Kit (Vazyme, Nanjing, China). Quantitative real-time PCR (qRT-PCR) was subsequently performed using 2 × SYBR Green PCR Master Mix (Thermo Fisher Scientific) with a real-time PCR system. Absolute quantification was conducted using standard curves generated from in vitro transcribed RNA or plasmid DNA with known copy numbers. Data were normalized to GAPDH expression levels and analyzed via the 2^−ΔΔCt^ method for relative quantification. Each sample was analyzed in triplicate, and technical replicates exhibiting Ct variations exceeding 0.5 were excluded from the analysis.

### 2.7. Determination of the Effect of PB2-E343K Mutation on PB2 Structure

Structural models of PB2 protein were built by homology-based modeling using SWISS-MODEL software (https://swissmodel.expasy.org (access on 23 December 2024)), using the crystal structures as described previously [15]. Structural images were generated using PyMOL 3.0 (The PyMOL molecular graphics system). The amino acid sequences of PB2-343E and PB2-343K protein were entered into the RoseTTAFold 2 server, and the 343 site structure of PB2 protein was predicted, as described previously [16]. The protein structure figures of the wild-type and mutant proteins were generated with the PyMol visualization software, version 2.5.0 (The PyMOL Molecular Graphics System). To assess the stability of the PB2 protein during simulations, we calculated the topological domain-specific root-mean-square deviation (RMSD) relative to the initial structures.

### 2.8. Statistical Analysis

Statistical significance was determined using Prism (GraphPad Software 8.0, San Diego, CA, USA) with an analysis of variance (ANOVA). A *p*-value less than 0.05 was regarded as statistically significant, whereas a *p*-value less than 0.01 was considered highly statistically significant. Each assay was carried out as a minimum of three independent experiments. The standard deviation is represented by the error bars.

## 3. Results

### 3.1. The Lysine at Position 343 in the PB2 Gene of the CQ445 Virus Attenuates the CQ91 Virus in Mice

During routine surveillance, we isolated two H1N1 viruses, CQ91 and CQ445, from pigs. Genomic sequence analysis revealed that these viruses share similar genetic compositions. Specifically, their hemagglutinin (HA), neuraminidase (NA), and matrix (M) genes are derived from EA H1N1 viral lineages. The PB2, PB1, PA, and NP genes are from 2009/H1N1 viruses, and the NS genes are from triple-reassortment H1N2 viruses. The sequences of the two viruses are available in the Global Initiative on Sharing Avian Influenza Data database (CQ91 accession numbers EPI2046587 to EPI2046591, EPI2046593 to EPI2046595; CQ445 accession numbers EPI2046531 to EPI2046535, EPI2046537 to EPI2046539). These two viruses exhibit an identical gene constellation to the G4 viruses described by Meng et al. [4]. We evaluated the replication kinetics and virulence phenotypes of the influenza viruses CQ91 and CQ445 in BALB/c mice. While both viruses demonstrated replicative capacity in mouse nasal turbinates and lungs, they exhibited significant differences in pathogenicity, with CQ91 showing greater virulence (MLD_50_ = 5.4 log_10_ EID_50_) compared to CQ445 (MLD_50_ = 6.6 log_10_ EID_50_) (Figure 1). Whole-genome sequencing revealed four amino acid variations between the two variants across three proteins: HA, PB2, and NP. Specifically, these variations involve a serine (S)-to-alanine (A) substitution at position 138 of the HA protein (H3 numbering), a glutamine (Q)-to-proline (P) mutation at position 20 of the NP protein, a phenylalanine (F)-to-serine (S) mutation at position 438 of the NP protein, and a glutamate (E)-to-lysine (K) substitution at position 343 of the PB2 protein. Notably, the amino acids at these positions in CQ91 are S, Q, F, and E, respectively, whereas those in CQ445 are A, P, S, and K. To identify the genes contributing to the pathogenicity difference in the two viruses in mice, we first established an eight-plasmid reverse genetics system. Using the previously reported strategy [17], we rescued the two viruses by transfecting constructed plasmids into permissive cell lines, followed by sequencing to confirm genome integrity and plaque assays to assess virus viability. Mouse studies were then conducted to evaluate the rescued viruses, rCQ91 and rCQ445, with strict experimental controls. As shown in Figure 1, both rCQ91 and rCQ445 had the same MLD_50_ as their wild-type parental viruses, validating our system and strategy and ensuring that subsequent pathogenicity differences could be attributed to genetic variations.

To investigate the genetic determinants underlying the pathogenicity differences between the two viruses, we conducted murine pathogenicity assays with CQ91 and its gene-reassortant variants. In the CQ91 genetic background, reassortant viruses harboring either the HA or NP gene of CQ445 displayed virulence comparable to the rCQ91 virus, with an MLD_50_ of 5.4 log_10_ EID_50_. By contrast, the reassortant carrying the PB2 gene of CQ445 (rCQ91-CQ445PB2) showed a 15.8-fold attenuation in mice relative to rCQ91, with MLD_50_ values of 6.6 versus 5.4 log_10_ EID_50_ (Figure 1 and Figure 2A). These results demonstrate that PB2 plays an essential role in the virulence of CQ91 in mice. To further validate the functional role of this genomic segment, we performed complementation experiments using the rCQ445 virus. In the CQ445 genetic background, reassortant viruses harboring the PB2 gene of CQ91 displayed a virulence comparable to the rCQ91 virus, with an MLD_50_ of 5.4 log_10_ EID_50_. In contrast, the reassortant carrying the PB2 gene of CQ91 (rCQ445-CQ91PB2) showed a 15.8-fold enhancement in virulence relative to rCQ445, with MLD_50_ values of 5.4 versus 6.6 log_10_ EID_50_ (Figure 1 and Figure 2A). These two sets of mutually validating experimental results demonstrate that the PB2 gene plays a pivotal role in regulating viral pathogenicity and replication capacity. Additionally, viral titers in the lungs of mice infected with rCQ91-CQ445PB2 were significantly lower than those in the corresponding lungs of rCQ91-infected mice, as documented in Figure 2B. Between the two viruses, a single amino acid substitution from E to K occurs at PB2 position 343. Concomitantly, the E at PB2 343 in CQ91 confers enhanced virulence to the CQ445 virus in murine models, whereas the K residue at this position in CQ445 mediates the attenuation of CQ91 virulence. In all the pathogenicity tests on mice, no virus was detected in brain, spleen and kidney tissues. Pathological studies indicated that most of the lungs showed mild damage after infection with rCQ445 and rCQ91-CQ445PB2 (Figure 2C). By contrast, the lungs of the CQ91 virus-infected mice showed extensive damage with the loss of the normal alveolar architecture, and a more viral antigen was detected in the epithelial cells of the bronchus and alveoli (Figure 2D). These results collectively highlight the critical role of the PB2 gene, particularly the amino acid at position 343, in modulating the virulence of CQ91 in the mouse model.

### 3.2. The Amino Acid at Position 343 in the PB2 Protein Affects Viral Replication in MDCK and A549 Cells

To investigate the impact of the PB2-K343E mutation on the viral replicative capacity in vitro, the multicycle growth levels of rCQ91, rCQ445, rCQ91-445PB2, and rCQ445-CQ91PB2 were compared in MDCK and A549 cells. As shown in Figure 3A (A549 cells) and Figure 3B (MDCK cells), the rCQ91 virus exhibited a more rapid growth pattern than the rCQ445 virus. Notably, the titers of rCQ445-CQ91PB2 were significantly higher than those of rCQ445, indicating enhanced replication. Conversely, the titers of rCQ91-CQ445PB2 were significantly lower than both rCQ91 and rCQ445-CQ91PB2. These findings align with the observed replication and virulence profiles of these viruses in mice. Collectively, they demonstrate that the E343K amino acid mutation in PB2 impairs the replication of swine H1N1 virus in mammalian cells. Conversely, introducing 343E into the PB2 of the CQ445 virus significantly enhances its growth ability at 37 °C.

### 3.3. PB2-K343E Mutation Increases CQ445 Polymerase Activity in 293T Cells

The PB2 protein, a key component of the viral ribonucleoprotein (RNP) complex, is central to influenza virus replication and virulence, with its functions intricately linked to viral transcription and genome replication [18,19]. The RNP complex, comprising PB2, PB1, PA, and NP, forms the enzymatic core responsible for viral RNA synthesis within host cells. To characterize functional differences between CQ91 and CQ445, we employed a luciferase-based minigenome assay in 293T cells, as previously described [9]. This assay measures viral polymerase activity by quantifying reporter gene expression driven by the influenza virus promoter, enabling a precise comparison of transcriptional efficiency. Significantly, the CQ91 RNP complex exhibited over 2-fold higher polymerase activity than CQ445 (Figure 4). To investigate the role of PB2 in this discrepancy, we constructed chimeric RNP complexes through reciprocal PB2 gene exchange. Replacing PB2 in CQ91 with its ortholog from CQ445 significantly reduced polymerase activity, whereas introducing CQ91-PB2 into CQ445 substantially enhanced this activity. These reciprocal experiments confirmed that PB2 serves as a primary determinant of the observed activity difference. Together, these results suggest that the PB2-K343E mutation enhances the polymerase activity of CQ445 in 293T cells.

### 3.4. The Amino Acid at Position 343 in PB2 Affects the Transcription Stage of the Viral Replication Process

During the influenza viral life cycle, the FluPol complex—composed of PB1, PB2, and PA—mediates the transcription of viral genomic RNA (vRNA) into messenger RNA (mRNA) and the replication of vRNA through a complementary RNA (cRNA) intermediate. As a critical subunit of the viral ribonucleoprotein (vRNP) complex, PB2 is essential in both vRNA replication and mRNA transcription. To determine whether the PB2 amino acid substitution at position 343 affects viral replication or transcription, we reconstituted vRNP complexes in 293T cells. This was achieved by transfecting wild-type or mutant polymerase (PB1, PB2, PA) and nucleoprotein (NP) expression plasmids, along with a vRNA template plasmid, to generate viral mRNA, cRNA intermediates, and genomic vRNA. Twenty-four hours post-transfection, qRT-PCR was used to quantify luciferase-specific cRNA, vRNA, and mRNA levels, enabling the analysis of replication and transcriptional activity. The results showed that the mRNA levels in cells transfected with the rCQ91-CQ445PB2 mutant plasmid were significantly lower than those in cells transfected with the rCQ91 wild-type plasmid, while the mRNA levels in cells transfected with the rCQ445-CQ91PB2 mutant plasmid were significantly higher than those in cells transfected with the rCQ445 wild-type plasmid (Figure 5C). However, the levels of vRNA and cRNA showed no significant differences across the four groups (Figure 5A,B). Thus, the findings indicate that the amino acid at PB2 position 343 impacts the transcriptional stage of viral replication.

### 3.5. The Three-Dimensional Structure of the PB2-K343E Protein

The functionality of active sites, binding sites, and catalytic sites within viral proteins is intricately tied to the precise three-dimensional (3D) arrangement of their structures. In this study, we conducted a comparative analysis of the 3D structures of the CQ91-PB2-343E and CQ445-PB2-343K proteins.

As depicted in the figure, the right-hand side, colored in green, represents the PB2-343K variant, while the left-hand side, in blue, corresponds to PB2-343E. The 343 residue is specifically highlighted in pink. The amino acid substitution from E to K leads to a substantial alteration in the R group, transitioning from -(CH_2_)_2_-COOH to -(CH_2_)_4_-NH_3_. This modification has the potential to influence the relative positions and conformations of neighboring amino acid residues, consequently giving rise to a certain steric hindrance effect. Moreover, given that E is negatively charged and K is positively charged, the E343K mutation can also exert an impact on the protein’s stability, folding, and functionality, particularly in its interactions with molecules of opposite charge (Figure 6A).

Regarding the impact of the simulated PB2-E343K mutation on the PB2 structure, the root-mean-square deviation (RMSD) value was calculated to be 0.005 following the mutation of residue E to K. This negligible RMSD value indicates that there are no significant structural changes in the PB2 protein. The minimal difference between the original (343E) and the mutated (343K) structures strongly suggests that the E343K mutation is unlikely to cause substantial alterations to the overall architecture of the protein (Figure 6B).

### 3.6. PB2 Amino Acid 343E Is Dominant in H1 and H3 Subtypes of Human and Swine Influenza Viruses

Finally, we determined the presence of the PB2 residue 343E among various influenza A virus isolates. Over 136,149 amino acid sequence records for the full-length PB2 of human and swine influenza viruses of the H1 and H3 subtypes were available in the GISAID database between January 2018 and September 2024. PB2-343E was almost completely conserved among the isolates (≥99.9%); only four isolates had residue 343K (Table 1). The PB2-343K mutation is rarely found among human and swine isolates, and it is logical to presume that it could not have conferred a selective advantage in mammalian adaptation.

## 4. Discussion

The H1N1 subtype SIV is extensively prevalent across diverse regions of China. It occasionally undergoes host jumps and reassortment events with other subtypes of influenza viruses, thereby posing a substantial threat to public health [4,20,21,22]. Over two thirds of the tested EA H1N1 viruses showed weak or no reactivity with serum antibodies induced by the 2009/H1N1 vaccine [4]. This suggests that pre-existing human immunity might be inadequate to prevent the infection and transmission of EA H1N1 viruses currently circulating in pig populations. Over the last several decades, there have been dozens of documented cases of human infections with EA H1N1 viruses [4,22,23,24]. The emergence of novel virus strains with the potential to trigger human epidemics or pandemics is a highly probable scenario. As such, it is crucial to conduct comprehensive investigations into mutations in swine viruses that could enhance their pathogenicity or transmissibility in human and other mammalian hosts. In this research, two EA H1N1 swine influenza viruses, namely, CQ91 and CQ445, were successfully isolated. Despite their genetic similarity, these viruses exhibited remarkably distinct pathogenicity levels in mice. Through reverse genetics approaches, the PB2 gene was identified as a key determinant of the virulence disparity between the two viruses in murine models. Notably, the E343K mutation within the PB2 gene was found to substantially reduce the virulence of CQ91 in mice. Further investigations revealed that the E343K mutation in PB2 induced a decrease in viral polymerase activity, which ultimately contributed to the attenuation of the H1N1 virus’s virulence in mice.

The activity of the influenza virus polymerase has a crucial impact on the replication of the influenza virus. For the 2009/H1N1 virus, the PB2 271A mutation has been shown to enhance viral polymerase activity and promote viral growth within mammalian cells [25,26]. Additionally, research conducted in cultured cells has revealed that the 590/591 (SR) polymorphism in the PB2 protein enables the 2009/H1N1 virus to circumvent host restrictions by enhancing its polymerase activity [27]. The PB2 protein is a key determinant of the viral host range, with characteristic markers 627K and 701N recognized as critical in defining the host range and serving as indicators of pathogenicity and transmissibility [18,28]. Notably, 627K and 701N in the PB2 protein are not universally applicable molecular markers for enhancing mammalian pathogenicity in influenza viruses. For 2009/H1N1-like viruses, studies have shown that influenza viruses harboring the 627K and 701N mutations within the 2009 H1N1 pandemic virus polymerase complex significantly exhibit attenuation in cell culture and murine models [29]. Therefore, it is imperative to identify additional molecular markers for 2009/H1N1 viruses to unravel their mechanisms of adaptation to mammalian hosts.

From a structural biology perspective, amino acid substitutions in viral proteins can induce far-reaching consequences beyond mere structural or charge changes. These alterations often perturb the delicate balance of protein dynamics, intermolecular interactions, and functional outputs. The gene transcription of IAV initiates from the process of snatching the cap structure (cap-snatching) from the host mRNA. The cap-binding domain of the PB2 protein is responsible for interacting with the 5′-end cap structure of the host pre-messenger ribonucleic acid (pre-mRNA). The crystal structure and functional characteristics of the cap-binding domain of the PB2 protein have become a widely studied topic [30,31,32]. The amino acid sequence of a protein directly dictates its functional properties. Beyond structural changes, amino acid substitutions can alter electrostatic interactions with other molecules. The impact of electrostatic interactions is crucial in determining the structural integrity of proteins, the way they fold, their binding capabilities, the process of condensation, and various biological functions [33]. Sequence analysis revealed that residue 343 of the PB2 protein resides within the cap-binding domain, a critical region for viral transcription. After confirming that the amino acid at position PB2-343 influences viral transcriptional activity, we investigated its role in cap snatching, a process central to host–virus interactions, specifically examining whether PB2-343 modulates influenza virus binding to host mRNA caps through conformational changes in the PB2 protein. The comparative structural modeling of CQ91-PB2-343E and CQ445-PB2-343K showed no significant conformational alterations induced by the E343K substitution, suggesting that structural remodeling is not the primary mechanism underlying functional divergence.

Notably, E and K exhibit distinct biochemical properties: E is an acidic residue, while K is basic, creating a charge disparity at position 343. Additionally, the K side chain is approximately 50% longer and more sterically bulky than E, potentially altering interactions with host cap-binding proteins. We hypothesize that the E343K mutation modulates cap-snatching efficiency through two non-mutually exclusive mechanisms: (1) the disruption of electrostatic interactions with host cap structures via charge alterations and (2) steric hindrance affecting the accessibility of the cap-binding pocket. These hypotheses require experimental validation, such as cap-binding affinity assays. This mechanistic inference underscores the importance of amino acid polarity and steric properties in viral host adaptation, providing a framework for future investigations into how the amino acid residue at position 343 of the PB2 protein influences influenza virus transcription and pathogenicity.

## 5. Conclusions

In conclusion, our study demonstrates that the lysine at position 343 in PB2 of the CQ445 virus attenuates the CQ91 virus in mice by impairing viral replication in cells, reducing polymerase activity, and disrupting viral mRNA transcription. Although the E343K mutation induces minimal structural changes in PB2, its rarity in human and swine influenza viruses suggests that it confers no selective advantage in mammalian adaptation. This study enriches influenza research theories and provides insights for developing live attenuated vaccines and antiviral drugs.

## Figures and Tables

**Figure 1 viruses-17-01018-f001:**
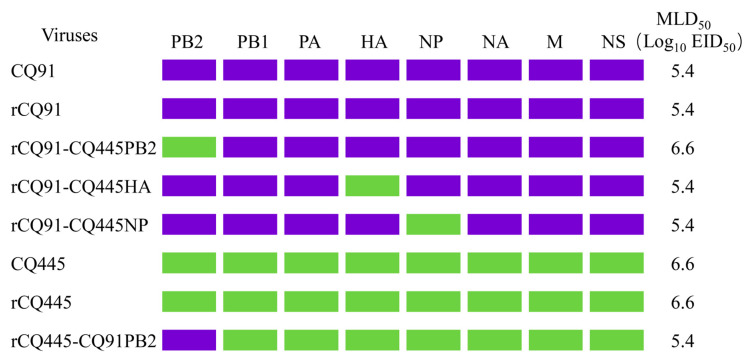
The 50% mouse lethal dose (MLD_50_) of rescued reassortant viruses. Viral gene segments are denoted by colored bars. Specifically, segments originating from CQ91 and CQ445 are represented in blue and green, respectively. The MLD_50_ of the rescued viruses was measured following the procedures detailed in the Materials and Methods section.

**Figure 2 viruses-17-01018-f002:**
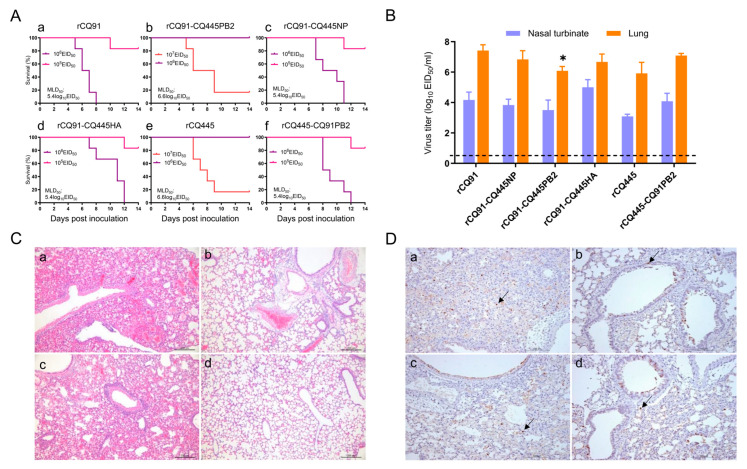
Replication and pathogenicity of rCQ91, rCQ445, and their reassortant viruses in mice. (**A**) Six mice per group were inoculated intranasally with the indicated doses of the six viruses and monitored for the survival rate of mice (a–f) for 14 days. Kaplan–Meier analysis and log-rank tests were used for survival estimation and group comparisons. (**B**) BALB/c mice were inoculated intranasally with 10^6^EID_50_ of rCQ91, rCQ445, and their reassortants. Three mice per group were euthanized on day 3 post infection. Lungs and nasal turbinates were collected for virus titrations in eggs. The data shown comprise means ± SDs. * *p* < 0.05 compared with the lung virus titer values of rCQ91. (**C**) Histopathologic changes in lungs caused by rCQ91, rCQ445, rCQ445-CQ91PB2, and rCQ91-CQ445PB2 (a, b, c, and d) in mice. The lungs of mice inoculated with 6.0 log_10_ EID_50_ of CQ91 and rCQ445-CQ91PB2 viruses show severe pathological lesions (a and c, respectively). The lungs of mice inoculated with 6.0 log_10_ EID_50_ of rCQ445 and rCQ91-CQ445PB2 viruses show mild pathological lesions. Images (a–d), bar = 200 μm. (**D**) The viral antigen (arrow) was mainly detected in the epithelial cells of the bronchus by means of immunohistochemical staining in the lungs of rCQ91, rCQ445, rCQ445-CQ91PB2, and rCQ91-CQ445PB2 virus-infected mice (a, b, c, and d). Images (a–d), bar = 100 μm.

**Figure 3 viruses-17-01018-f003:**
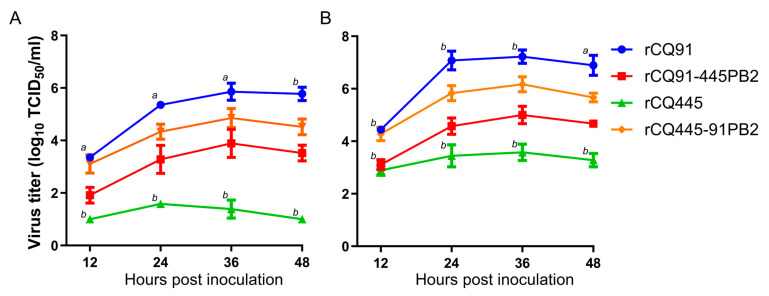
Multistep growth of rescued viruses in cultured cells. A549 (**A**) or MDCK (**B**) cells were infected with the specified viruses at a multiplicity of infection (MOI) of 0.01 and cultivated at 37 °C in the presence of 1 μg/mL trypsin. Culture supernatants were collected at the designated time points, and virus titers were quantified. The data presented represent the mean ± standard deviation (SD) from three independent experiments. At each time point, data were analyzed by two-way ANOVA followed by Dennett’s test, comparing each group between rCQ91 and rCQ91-445PB2 and between rCQ445 and rCQ445-91PB2 (*a*, *p* < 0.05; *b*, *p* < 0.01).

**Figure 4 viruses-17-01018-f004:**
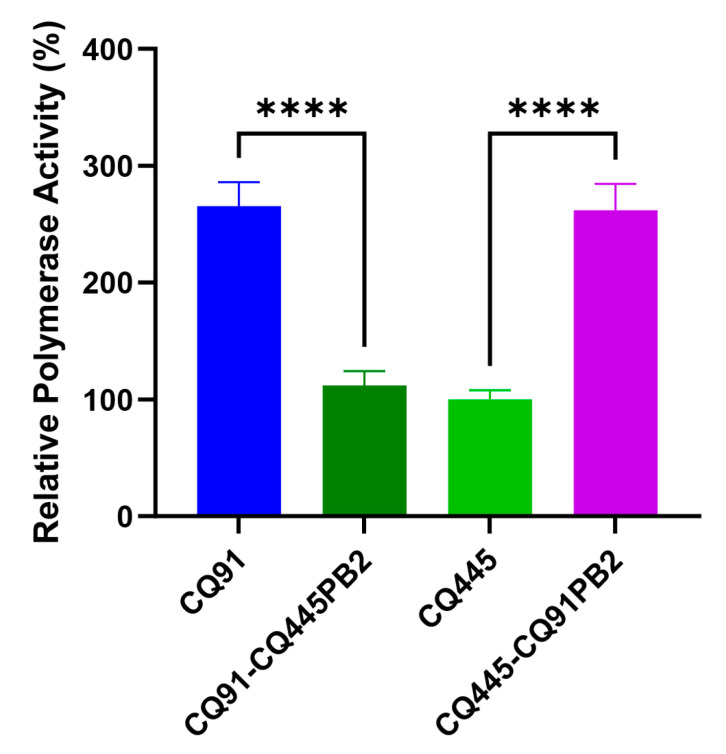
Polymerase activity of wild-type and various mutant polymerase complexes; 293T cells were transfected with plasmids expressing pCAGGS-PB1, pCAGGS-PB2 (wild-type or mutant), pCAGGS-PA, pCAGGS-NP of CQ91 or CQ445, and pPolI-NP-Luci and pRL-TK. Polymerase activity was measured at 24 hpt. All the data are presented as the means ± SDs, and comparisons were performed using one-way ANOVA. **** *p* < 0.0001 compared with the values of the vRNP of rCQ445; **** *p* < 0.0001 compared with the values of the vRNP of rCQ91. Data shown are the mean  ±  SD from three independent experiments.

**Figure 5 viruses-17-01018-f005:**
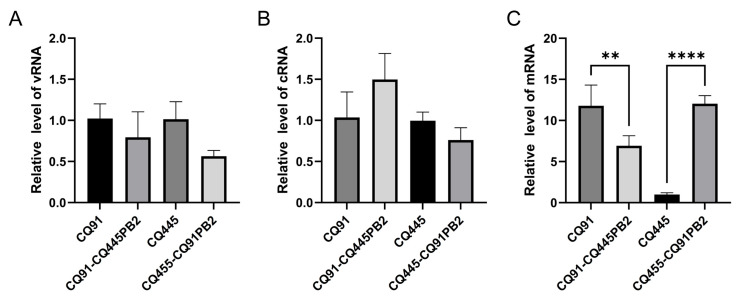
Quantitative real-time PCR results for luciferase vRNA, cRNA, and mRNA in HEK293T cells. (**A**–**C**) The pCAGGS-PB1, pCAGGS-PB2 (wild-type or mutant), pCAGGS-PA, pCAGGS-NP, and pPolI-vRNA plasmids were mixed and transfected into HEK293T cells. RNA was isolated from cells after 24 h of incubation and analyzed by a primer extension assay as described in Materials and Methods. (**A**) The vRNA, (**B**) cRNA, and (**C**) mRNA level of the luciferase gene. All the data are presented as the means ± SDs, and comparisons were performed using one-way ANOVA. ** *p* < 0.01, **** *p* < 0.0001. The results are from at least three different experiments.

**Figure 6 viruses-17-01018-f006:**
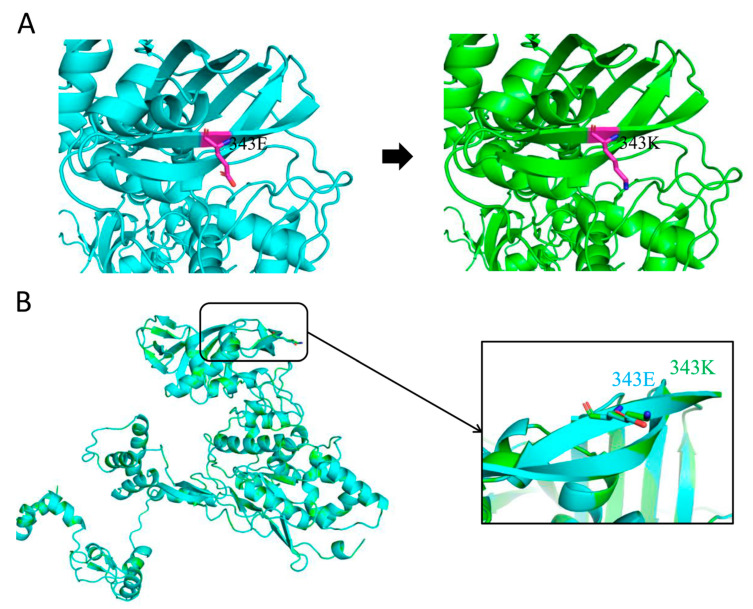
The three-dimensional structure of the PB2-343E/K protein. (**A**) The overall 3D structure analysis of PB2; modeling was performed with Swiss-Model, and images were superimposed using PyMOL. The PB2-343E protein is shown in blue, and the PB2-343K protein is shown in green. (**B**) The root-mean-square deviation (RMSD) was analyzed for the PB2-343E and PB2-343K proteins. The RoseTTAFold modeling server was used to predict the protein structures of PB2-343E and PB2-343K based on their amino acid sequences. The protein structure figures were generated with PyMol visualization software. A root-mean-square deviation (RMSD) value below 1 indicates that the mimic modification mutations in PB2 do not substantially alter the protein’s structure.

**Table 1 viruses-17-01018-t001:** Amino acid distribution characteristics of PB2-343 in influenza viruses isolated from different hosts between January 2018 and September 2024.

	PB2-343E	PB2-343K	PB2-343 Other Residues	Total Number
Human H1N1	54,348	3	38	54,389
Human H3N2	77,007	0	28	77,035
Swine H1N1	1858	1	0	1849
Swine H1N2	1620	0	0	1620
Swine H3N2	1256	0	0	1256
Total	136,089	4	66	136,149

## Data Availability

The data is available upon request.

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
