# Peer review of "Glutamic Acid at Position 343 in PB2 Contributes to the Virulence of H1N1 Swine Influenza Virus in Mice"

_viruses, 2025, doi:10.3390/v17071018_

Round 1

Reviewer 1 Report

Comments and Suggestions for Authors

This manuscript is very important for understanding the functional activity of different mutation (aminoacid substitution) in influenza viruses. Becouse the influenza viruses have the fragmental genome they are particullary changeble. The role of most number of mutation  the genome of influenza viruses still unstudied. This article clarified the role of K343E aminoacid substitution in PB2 swine influenza A(H1N1)virus for increasing the virulence and replication rate of this virus. These facts confirmed by authors in many experiments with using different laboratory models (cells and mice). The suggestions for authors - continuation these experiments with another influenza viruses for investigation of role the different mutations in the genome of influenza viruses.

One suggestion: Figure 4 look like disproportionately according the scale of other figures. Can be improved.

Author Response

Comments1: This manuscript is very important for understanding the functional activity of different mutation (aminoacid substitution) in influenza viruses. Becouse the influenza viruses have the fragmental genome they are particullary changeble. The role of most number of mutation  the genome of influenza viruses still unstudied. This article clarified the role of K343E aminoacid substitution in PB2 swine influenza A(H1N1)virus for increasing the virulence and replication rate of this virus. These facts confirmed by authors in many experiments with using different laboratory models (cells and mice). The suggestions for authors - continuation these experiments with another influenza viruses for investigation of role the different mutations in the genome of influenza viruses.

One suggestion: Figure 4 look like disproportionately according the scale of other figures. Can be improved.

Response 1: Thank you very much for your valuable comments and suggestions on our manuscript. We greatly appreciate your recognition of the significance of our work regarding the K343E amino acid substitution in the PB2 of swine influenza A(H1N1) virus.

We have carefully noted your suggestion about Figure 4. We agree that there was a disproportion in its scale compared to other figures, and we have already revised it to ensure better consistency with the rest of the figures.

We also highly value your proposal to extend the experiments to other influenza viruses to investigate the roles of different mutations, and we will seriously consider this direction for our future research.

Thank you again for your time and input, which have helped improve our manuscript.

Reviewer 2 Report

Comments and Suggestions for Authors

The manuscript by Yanwen Wang et al. presents a study of two H1N1 swine influenza virus strains, CQ91 and CQ445, isolated from pigs in China, which exhibit different virulence in mice despite having similar genomic constellations. They identify a single amino acid substitution—E343K in the PB2 protein—as the key determinant responsible for the attenuation of CQ445. Using reverse genetics, the authors demonstrate that introducing the PB2-343K mutation into the more virulent CQ91 strain significantly reduces its pathogenicity and replication efficiency in mice, while restoring the 343E residue in CQ445 enhances virulence. In vitro experiments further show that the E343K substitution impairs viral replication in MDCK and A549 cells and decreases polymerase activity in minigenome assays. Mechanistically, the mutation affects the transcriptional stage of the viral replication cycle. Structural modeling suggests that the charge reversal introduced by the E343K mutation alters local electrostatic interactions without major conformational shifts. The authors also report that PB2-343E is highly conserved among human and swine H1/H3 influenza viruses, indicating a strong adaptive advantage. They conclude that PB2-343E is a critical determinant of influenza virus pathogenicity in mammals and plays a significant role in host adaptation.

Overall, the manuscript is well written, and the methods are clearly described. However, I believe there are a few critical issues in the study that weaken the authors' conclusions.

Major concerns:

  1. In the Results section (page 6, lines 147–150), the authors state: “Whole-genome sequencing revealed three amino acid variations between the variants across three proteins: HA, PB2, and NP. Specifically, these involve a serine (S) to alanine (A) substitution at position 138 of the HA protein (H3 numbering), a glutamine (Q) to proline (P) mutation at position 20 of the NP protein, and a glutamate (E) to lysine (K) substitution at position 343 of the PB2 protein”. However, this statement appears to be inaccurate. The amino acid (AA) differences between CQ91 and CQ445 are more than the 3 reported substitutions. There are 2 AA differences in the NP segment, including one reported by the authors at position 20, and an additional one at position 438 (S/F). There is one AA difference in the NA segment at position 2 (N/D) that the authors did not report. These discrepancies suggest that the genomic comparison may not have been comprehensive. Please see the alignments of the NP and NA segments of CQ445 and CQ91 at the end of my review.
  2. The authors constructed several chimeric viruses, including rCQ91-CQ445PB2, rCQ91-CQ445HA, rCQ91-CQ445NP, and rCQ445-CQ91PB2, and assessed their MLD50 values, viral titers, survival rates, histopathological changes, and viral antigen distribution in a mouse model (Figures 1 and 2). While these experiments are informative, I believe the study should include additional chimeric viruses, specifically rCQ445-CQ91HA and rCQ445-CQ91NP, to fully assess the contribution of the HA and NP segments from CQ91 in the attenuated CQ445 background. In fact, it appears that the authors may already have generated the rCQ445-CQ91NP construct, as they present polymerase activity data for this segment in Figure 4 (page 10).
  3. Figure 2B (page 7) presents survival data for mice infected with the various rescued viruses. The results clearly show that rCQ91-CQ445PB2 significantly attenuates virulence and improves survival, while rCQ445-CQ91PB2 markedly increases mortality, supporting the central role of PB2-343 in pathogenicity. However, the data also indicate that rCQ91-CQ445NP completely rescues the mice and that rCQ91-CQ445HA provides partial protection. These findings suggest that the NP and HA segments may also contribute to viral attenuation. The authors should address and discuss these observations in the manuscript, as they appear to reveal additional genetic determinants influencing virulence in this model.
  4. Finally, the authors report that the PB2-343K variant is extremely rare (<0.1%) among human and swine H1 and H3 influenza viruses, as shown in Table 1 (page 14). This raises an important question regarding the origin of the 343K mutation in the CQ445 strain. Could this substitution have arisen during viral passage in the laboratory, or might it be the result of a sequencing artifact? The authors should clarify how the original CQ445 sequence was obtained, including details about the number of passages, the passage system used (e.g., eggs, cells), and whether sequencing was performed directly from the original isolate or from passaged virus. Addressing this point is important to rule out the possibility that PB2-343K is an experimental artifact rather than a naturally occurring variant.

Some minor concerns:

  1. On page 8, lines 170 to 172, the manuscript states to "All the data are presented as the means ± SDs ... three independent experiments". It seems to apply specifically to quantitative data shown in Figure 2A, not to Figure 2B.

  1. On page 8, lines 182 to 183, the author state “Additionally, viral titers in organs of mice infected with rCQ91-CQ445PB2 were significantly lower than those in corresponding organs of rCQ91-infected mice, as documented in Figure 2A.” However, Figure 2A presents viral titers only from lung tissue. It is unclear whether titers were measured in any other organs? The pathogenicity assessment (page 8, line 188) actually includes histopathological analysis of brain, spleen, and kidney.
  2. On page 11, lines 242 to 245, it states “The results showed that the mRNA levels in cells transfected with the rCQ91-CQ445PB2 mutant plasmid were significantly lower than those in cells transfected with the rCQ91 wild-type plasmid, while the mRNA levels in cells transfected with the rCQ445-CQ91PB2 mutant plasmid were significantly higher than those in cells transfected with the rCQ445 wild-type plasmid (Figure 5A)”. The mRNA quantification data are actually shown in Figure 5C, not Figure 5A.

(Alignments are formatted better in my attached file).

RID: 6CWHSVKV114     

Job Title:EPI2046527|NP|A/swine/Chongqing/445/2018|EPI_ISL_12941008|A_/_H1N1                         

Program: BLASTP

Query: EPI2046527|NP|A/swine/Chongqing/445/2018|EPI_ISL_12941008|A_/_H1N1 ID: lcl|Query_2218645(amino acid) Length: 498

Subject:EPI2046587|NP|A/swine/Chongqing/91/2018|EPI_ISL_12941067|A_/_H1N1 ID: lcl|Query_2218647(amino acid) Length: 498

Sequences producing significant alignments:

                                                                  Scientific      Common                     Max    Total Query   E   Per.   Acc.                       

Description                                                       Name            Name            Taxid      Score  Score cover Value Ident  Len        Accession       

EPI2046587|NP|A/swine/Chongqing/91/2018|EPI_ISL_12941067|A_/_H1N1                                 0          1029   1029  100%  0.0   99.60  498        Query_2218647   

Alignments:

>EPI2046587|NP|A/swine/Chongqing/91/2018|EPI_ISL_12941067|A_/_H1N1

Sequence ID: Query_2218647 Length: 498

Range 1: 1 to 498

Score:1029 bits(2661), Expect:0.0,

Method:Compositional matrix adjust.,

Identities:496/498(99%), Positives:496/498(99%), Gaps:0/498(0%)

Query  1    MASQGTKRSYEQMETGGERPDATEIRASVGRMIGGIGRFYIQMCTELKLSDYEGRLIQNS  60

            MASQGTKRSYEQMETGGER DATEIRASVGRMIGGIGRFYIQMCTELKLSDYEGRLIQNS

Sbjct  1    MASQGTKRSYEQMETGGERQDATEIRASVGRMIGGIGRFYIQMCTELKLSDYEGRLIQNS  60

Query  61   ITIERMVLSAFDERRNKYLEEHPSAGKDPKKTGGSIYRRIDGKWMRELILYDKEEIRRVW  120

            ITIERMVLSAFDERRNKYLEEHPSAGKDPKKTGGSIYRRIDGKWMRELILYDKEEIRRVW

Sbjct  61   ITIERMVLSAFDERRNKYLEEHPSAGKDPKKTGGSIYRRIDGKWMRELILYDKEEIRRVW  120

Query  121  RQANNGEDATAGLTHIMIWHSNLNDATYQRTRALVRTGMDPRMCSLMQGSTLPRRSGAAG  180

            RQANNGEDATAGLTHIMIWHSNLNDATYQRTRALVRTGMDPRMCSLMQGSTLPRRSGAAG

Sbjct  121  RQANNGEDATAGLTHIMIWHSNLNDATYQRTRALVRTGMDPRMCSLMQGSTLPRRSGAAG  180

Query  181  AAVKGIGTIAMELIRMIKRGINDRNFWRGENGRRTRVAYERMCNILKGKFQTAAQRAMMD  240

            AAVKGIGTIAMELIRMIKRGINDRNFWRGENGRRTRVAYERMCNILKGKFQTAAQRAMMD

Sbjct  181  AAVKGIGTIAMELIRMIKRGINDRNFWRGENGRRTRVAYERMCNILKGKFQTAAQRAMMD  240

Query  241  QVRESRNPGNAEIEDLIFLARSALILRGSVAHKSCLPACVYGLAVASGHDFEREGYSLVG  300

            QVRESRNPGNAEIEDLIFLARSALILRGSVAHKSCLPACVYGLAVASGHDFEREGYSLVG

Sbjct  241  QVRESRNPGNAEIEDLIFLARSALILRGSVAHKSCLPACVYGLAVASGHDFEREGYSLVG  300

Query  301  IDPFKLLQNSQVVSLIRPNENPAHKSQLVWMACHSAAFEDLRVSSFIRGKKVVPRGKLST  360

            IDPFKLLQNSQVVSLIRPNENPAHKSQLVWMACHSAAFEDLRVSSFIRGKKVVPRGKLST

Sbjct  301  IDPFKLLQNSQVVSLIRPNENPAHKSQLVWMACHSAAFEDLRVSSFIRGKKVVPRGKLST  360

Query  361  RGVQIASNENVETMDSNTLELRSRYWAIRTRSGGNTNQQKASAGQISVQPTFSVQRNLPF  420

            RGVQIASNENVETMDSNTLELRSRYWAIRTRSGGNTNQQKASAGQISVQPTFSVQRNLPF

Sbjct  361  RGVQIASNENVETMDSNTLELRSRYWAIRTRSGGNTNQQKASAGQISVQPTFSVQRNLPF  420

Query  421  ERATVMAAFSGNNEGRTSDMRTEVIRMMESAKPEDLSFQGRGVFELSDEKATNPIVPSFD  480

            ERATVMAAFSGNNEGRT DMRTEVIRMMESAKPEDLSFQGRGVFELSDEKATNPIVPSFD

Sbjct  421  ERATVMAAFSGNNEGRTFDMRTEVIRMMESAKPEDLSFQGRGVFELSDEKATNPIVPSFD  480

Query  481  MSNEGSYFFGDNAEEYDS  498

            MSNEGSYFFGDNAEEYDS

Sbjct  481  MSNEGSYFFGDNAEEYDS  498

RID: 6CX8FM48114     

Job Title:EPI2046534|NA|A/swine/Chongqing/445/2018|EPI_ISL_12941008|A_/_H1N1                         

Program: BLASTP

Query: EPI2046534|NA|A/swine/Chongqing/445/2018|EPI_ISL_12941008|A_/_H1N1 ID: lcl|Query_2859327(amino acid) Length: 469

Subject:EPI2046594|NA|A/swine/Chongqing/91/2018|EPI_ISL_12941067|A_/_H1N1 ID: lcl|Query_2859329(amino acid) Length: 469

Sequences producing significant alignments:

                                                                  Scientific      Common                     Max    Total Query   E   Per.   Acc.                       

Description                                                       Name            Name            Taxid      Score  Score cover Value Ident  Len        Accession       

EPI2046594|NA|A/swine/Chongqing/91/2018|EPI_ISL_12941067|A_/_H1N1                                 0          960    960   100%  0.0   99.79  469        Query_2859329   

Alignments:

>EPI2046594|NA|A/swine/Chongqing/91/2018|EPI_ISL_12941067|A_/_H1N1

Sequence ID: Query_2859329 Length: 469

Range 1: 1 to 469

Score:960 bits(2482), Expect:0.0,

Method:Compositional matrix adjust.,

Identities:468/469(99%), Positives:469/469(100%), Gaps:0/469(0%)

Query  1    MNPNQKIITIGSICMTIGIASLILQIGNIISIWVSHSIQIENQNQSEICNQNVITYENNT  60

            M+PNQKIITIGSICMTIGIASLILQIGNIISIWVSHSIQIENQNQSEICNQNVITYENNT

Sbjct  1    MDPNQKIITIGSICMTIGIASLILQIGNIISIWVSHSIQIENQNQSEICNQNVITYENNT  60

Query  61   WVNQTYVNVSNTNFVAEQVVASVKLAGNSSLCPVSGWAIYSKDNSVRIGSKGDVFVIREP  120

            WVNQTYVNVSNTNFVAEQVVASVKLAGNSSLCPVSGWAIYSKDNSVRIGSKGDVFVIREP

Sbjct  61   WVNQTYVNVSNTNFVAEQVVASVKLAGNSSLCPVSGWAIYSKDNSVRIGSKGDVFVIREP  120

Query  121  FISCSHLECRTFFLTQGALLNDKHSNGTIKDRSPYRTLMSCPIGEVPSPYNSRFESVAWS  180

            FISCSHLECRTFFLTQGALLNDKHSNGTIKDRSPYRTLMSCPIGEVPSPYNSRFESVAWS

Sbjct  121  FISCSHLECRTFFLTQGALLNDKHSNGTIKDRSPYRTLMSCPIGEVPSPYNSRFESVAWS  180

Query  181  ASACHDGTSWLTIGISGPDNGAVAVLKYNGIITDTIKSWRKNILRTQESECACVNGSCFT  240

            ASACHDGTSWLTIGISGPDNGAVAVLKYNGIITDTIKSWRKNILRTQESECACVNGSCFT

Sbjct  181  ASACHDGTSWLTIGISGPDNGAVAVLKYNGIITDTIKSWRKNILRTQESECACVNGSCFT  240

Query  241  VMTDGPSNGQASYKIFKIERGKVVKSVELNAPNYHYEECSCYPESSEIICVCRDNWHGSN  300

            VMTDGPSNGQASYKIFKIERGKVVKSVELNAPNYHYEECSCYPESSEIICVCRDNWHGSN

Sbjct  241  VMTDGPSNGQASYKIFKIERGKVVKSVELNAPNYHYEECSCYPESSEIICVCRDNWHGSN  300

Query  301  RPWVSFNQNLEYQIGYICSGIFGDNPRPDDKTGSCGPVFPNGSNGVKGFSFKYGNGVWIG  360

            RPWVSFNQNLEYQIGYICSGIFGDNPRPDDKTGSCGPVFPNGSNGVKGFSFKYGNGVWIG

Sbjct  301  RPWVSFNQNLEYQIGYICSGIFGDNPRPDDKTGSCGPVFPNGSNGVKGFSFKYGNGVWIG  360

Query  361  RTKSTSSRMGFEMIWDPDGWTRTDDKFSVKQDIIGITDWSGYSGSFVQHPELTGLDCMRP  420

            RTKSTSSRMGFEMIWDPDGWTRTDDKFSVKQDIIGITDWSGYSGSFVQHPELTGLDCMRP

Sbjct  361  RTKSTSSRMGFEMIWDPDGWTRTDDKFSVKQDIIGITDWSGYSGSFVQHPELTGLDCMRP  420

Query  421  CFWVELIRGRPKENTIWTSGSSISFCGVNSDTVGWSWPDGAELPFTIDK  469

            CFWVELIRGRPKENTIWTSGSSISFCGVNSDTVGWSWPDGAELPFTIDK

Sbjct  421  CFWVELIRGRPKENTIWTSGSSISFCGVNSDTVGWSWPDGAELPFTIDK  469     

Author Response

Reviewer 2:

The manuscript by Yanwen Wang et al. presents a study of two H1N1 swine influenza virus strains, CQ91 and CQ445, isolated from pigs in China, which exhibit different virulence in mice despite having similar genomic constellations. They identify a single amino acid substitution—E343K in the PB2 protein—as the key determinant responsible for the attenuation of CQ445. Using reverse genetics, the authors demonstrate that introducing the PB2-343K mutation into the more virulent CQ91 strain significantly reduces its pathogenicity and replication efficiency in mice, while restoring the 343E residue in CQ445 enhances virulence. In vitro experiments further show that the E343K substitution impairs viral replication in MDCK and A549 cells and decreases polymerase activity in minigenome assays. Mechanistically, the mutation affects the transcriptional stage of the viral replication cycle. Structural modeling suggests that the charge reversal introduced by the E343K mutation alters local electrostatic interactions without major conformational shifts. The authors also report that PB2-343E is highly conserved among human and swine H1/H3 influenza viruses, indicating a strong adaptive advantage. They conclude that PB2-343E is a critical determinant of influenza virus pathogenicity in mammals and plays a significant role in host adaptation.

Overall, the manuscript is well written, and the methods are clearly described. However, I believe there are a few critical issues in the study that weaken the authors' conclusions.

Major concerns:

Comments1: In the Results section (page 6, lines 147–150), the authors state: “Whole-genome sequencing revealed three amino acid variations between the variants across three proteins: HA, PB2, and NP. Specifically, these involve a serine (S) to alanine (A) substitution at position 138 of the HA protein (H3 numbering), a glutamine (Q) to proline (P) mutation at position 20 of the NP protein, and a glutamate (E) to lysine (K) substitution at position 343 of the PB2 protein”. However, this statement appears to be inaccurate. The amino acid (AA) differences between CQ91 and CQ445 are more than the 3 reported substitutions. There are 2 AA differences in the NP segment, including one reported by the authors at position 20, and an additional one at position 438 (S/F). There is one AA difference in the NA segment at position 2 (N/D) that the authors did not report. These discrepancies suggest that the genomic comparison may not have been comprehensive. Please see the alignments of the NP and NA segments of CQ445 and CQ91 at the end of my review.

Response 1: We appreciate the reviewer’s careful examination of our manuscript. We agree with the reviewer’s observation regarding the amino acid differences between the CQ91 and CQ445 variants. In the revised version, we have corrected the reported amino acid variations in the Results section (Page 6, Lines 148–153) to include the additional substitution in the NP segment (position 438, S/F). Regarding the NA segment, we rechecked the sequencing data and found no amino acid difference at position 2 (N/D) between the two variants. The alignment confirms that the NA sequences are identical, and we have double-checked and reuploaded the relevant sequencing files to ensure accuracy. We sincerely thank the reviewer for bringing these important details to our attention, which have helped improve the precision of our manuscript.

Comments 2: The authors constructed several chimeric viruses, including rCQ91-CQ445PB2, rCQ91-CQ445HA, rCQ91-CQ445NP, and rCQ445-CQ91PB2, and assessed their MLD50 values, viral titers, survival rates, histopathological changes, and viral antigen distribution in a mouse model (Figures 1 and 2). While these experiments are informative, I believe the study should include additional chimeric viruses, specifically rCQ445-CQ91HA and rCQ445-CQ91NP, to fully assess the contribution of the HA and NP segments from CQ91 in the attenuated CQ445 background. In fact, it appears that the authors may already have generated the rCQ445-CQ91NP construct, as they present polymerase activity data for this segment in Figure 4 (page 10).

Response 2: We sincerely appreciate this insightful suggestion. We apologize for any lack of clarity in our original manuscript and have now provided more detailed explanations on page 8, lines 182-192 of the revised version. Our experimental strategy was implemented in a stepwise manner: We initially introduced segments from the attenuated CQ445 strain into the virulent CQ91 background, which identified PB2 as the critical determinant of viral pathogenicity. We then performed reciprocal validation by introducing the CQ91 PB2 segment into the CQ445 background, confirming its pivotal role in virulence.

Regarding HA and NP segments: Our preliminary experiments demonstrated that replacing CQ91 HA or NP segments with their CQ445 counterparts did not significantly affect viral pathogenicity in mice. Therefore, we did not conduct reciprocal validation (introducing CQ91 HA/NP into CQ445 background) as these segments showed no association with virulence differences. We would like to clarify that the rCQ445-CQ91NP construct mentioned by the reviewer is not identical to the plasmid vector used in our in vitro polymerase activity assays (Figure 4), as they were designed for different experimental purposes. We hope this clarification better explains our experimental rationale. We are grateful for the reviewer's comment, which has helped us improve the manuscript's clarity. Thank you for your thoughtful review.

Comments 3: Figure 2B (page 7) presents survival data for mice infected with the various rescued viruses. The results clearly show that rCQ91-CQ445PB2 significantly attenuates virulence and improves survival, while rCQ445-CQ91PB2 markedly increases mortality, supporting the central role of PB2-343 in pathogenicity. However, the data also indicate that rCQ91-CQ445NP completely rescues the mice and that rCQ91-CQ445HA provides partial protection. These findings suggest that the NP and HA segments may also contribute to viral attenuation. The authors should address and discuss these observations in the manuscript, as they appear to reveal additional genetic determinants influencing virulence in this model.

Response 3: We sincerely appreciate the reviewer's careful observation. We apologize for any lack of clarity in our original presentation and have now revised Figure 2 to present the results more clearly and consistently with Figure 1. The MLD50 values for viruses with replaced NP or HA segments were identical to those of the wild-type virus, clearly demonstrating that these segments are not involved in viral attenuation. We have added detailed descriptions addressing this issue in the Results section (pages 182-192) to provide better context for interpreting these findings. We are grateful for your valuable comments, which have significantly improved the rigor of our data presentation.

Comments 4: Finally, the authors report that the PB2-343K variant is extremely rare (<0.1%) among human and swine H1 and H3 influenza viruses, as shown in Table 1 (page 14). This raises an important question regarding the origin of the 343K mutation in the CQ445 strain. Could this substitution have arisen during viral passage in the laboratory, or might it be the result of a sequencing artifact? The authors should clarify how the original CQ445 sequence was obtained, including details about the number of passages, the passage system used (e.g., eggs, cells), and whether sequencing was performed directly from the original isolate or from passaged virus. Addressing this point is important to rule out the possibility that PB2-343K is an experimental artifact rather than a naturally occurring variant.

Response 4: We sincerely appreciate the reviewer for raising this important question regarding the origin of the PB2-343K variant in the CQ445 strain. The CQ445 sample underwent one passage in chicken eggs prior to second-generation sequencing, which initially identified the 343K mutation. Before animal experiments, the virus was passaged once more in eggs and sequenced again, confirming the persistence of this mutation. Furthermore, post-animal experiment sequencing consistently detected the 343K mutation. We believe these results effectively exclude the possibility of an experimental artifact. We thank the reviewer for helping us strengthen the discussion of this critical issue.

Comments 5: On page 8, lines 170 to 172, the manuscript states to "All the data are presented as the means ± SDs ... three independent experiments". It seems to apply specifically to quantitative data shown in Figure 2A, not to Figure 2B.

Response 5: We agree with the reviewer’s suggestion and have revised the text accordingly. The statement now clearly specifies that the data presented as "means ± SDs" from "three independent experiments" apply specifically to the viral titer measurements, rather than the survival data. This clarification has been added on page 8, line 171-175 of the revised manuscript. Thank you for your careful reading and constructive feedback, which have helped improve the accuracy of our data presentation.

Comments 6: On page 8, lines 182 to 183, the author state “Additionally, viral titers in organs of mice infected with rCQ91-CQ445PB2 were significantly lower than those in corresponding organs of rCQ91-infected mice, as documented in Figure 2A.” However, Figure 2A presents viral titers only from lung tissue. It is unclear whether titers were measured in any other organs? The pathogenicity assessment (page 8, line 188) actually includes histopathological analysis of brain, spleen, and kidney.

Response 6: We sincerely appreciate the reviewer’s careful observation regarding viral titers in different organs. Indeed, we collected and attempted to detect viral titers in the spleen, kidney, and brain tissues. However, no virus was detected in these organs across all experimental groups (as noted in the revised manuscript, Page 9, lines 196-198). Since these organs tested negative, we chose not to include them in Figure 2 to emphasize the significant differences observed in the lung and nasal turbinate data. We appreciate this comment, as it has helped us improve the precision of our data presentation.

Comments 7: On page 11, lines 242 to 245, it states “The results showed that the mRNA levels in cells transfected with the rCQ91-CQ445PB2 mutant plasmid were significantly lower than those in cells transfected with the rCQ91 wild-type plasmid, while the mRNA levels in cells transfected with the rCQ445-CQ91PB2 mutant plasmid were significantly higher than those in cells transfected with the rCQ445 wild-type plasmid (Figure 5A)”. The mRNA quantification data are actually shown in Figure 5C, not Figure 5A.

Response 7: We sincerely appreciate the reviewer's careful reading of our manuscript. We acknowledge this citation error and have corrected the figure reference from "Figure 5A" to "Figure 5C" in the revised manuscript (page 11, line 247). This revision ensures accurate correspondence between the described mRNA quantification data and its presentation in the figures. Thank you for identifying this oversight, which has helped improve the precision of our manuscript.

Round 2

Reviewer 2 Report

Comments and Suggestions for Authors

All of my prior concerns have been addressed by the authors.